

# Development and application of a CRISPR/Cas12a-based reverse transcription–recombinase polymerase amplification assay with lateral flow dipstick and fluorescence detection for Getah virus

Boyang Xia[1], Ziyan Wang[2], Tiantian Fei[3], Yueyu Ma[4], Yaxi Guo[4], Dongliang Fei[4], Xiuwei Shu[5], Gang Zhao[6], Mingxiao Ma[2] and Hongxia Yuan[7]

[1] Collaborative Innovation Center for Zoonosis Prevention and Treatment of Jinzhou Medical University, Jinzhou, Liaoning, China
[2] College of Animal Husbandry and Veterinary Medicine, Jinzhou Medical University, Jinzhou, Liaoning, China
[3] Henan University of Chinese Medicine, Zhengzhou, Henan, China
[4] Experimental Animal Center, Jinzhou Medical University, Jinzhou, Liaoning, China
[5] Liaoning Yikang Biological Co., Ltd, Liaoyang, Liaoning, China
[6] Liaoning Provinsiale Landbou-ontwikkelingsdienssentrum, Shenyang, Liaoning, China
[7] The First Affiliated Hospital of Jinzhou Medical University, Jinzhou, Liaoning, China

Corresponding authors
Mingxiao Ma, lnjzmmx@163.com
Hongxia Yuan, womende20081018@163.com

## ABSTRACT

Getah virus (GETV), a mosquito-borne alphavirus classified as a zoonotic disease, primarily infects livestock, particularly pigs and horses. In recent years, it has re-emerged in multiple Asian countries, posing a potential threat to animal husbandry and public health. In this study, we developed a rapid and sensitive GETV detection method based on reverse transcription-recombinase polymerase amplification (RT-RPA) and the clustered regularly interspaced short palindromic repeats (CRISPR)/Cas12a system combined with a lateral flow dipstick (LFD) for visual readout. By leveraging sequence conservation in the GETV E2 envelope protein-coding regions, we engineered matched crRNA guides and amplification primers to develop a rapid CRISPR-Cas12a diagnostic workflow. The optimized platform combines RT-RPA (42 °C/20 min) with Cas12a's trans-nuclease activity, permitting multiplex detection via real-time fluorescence quantification or immunochromatographic strip visualization. Analytical evaluation demonstrated a detection capability of 10 copies/μL and exclusive specificity against four pathogen controls, including Japanese encephalitis virus and pseudorabies virus. Validation performed using simulated clinical samples revealed 100% concordance between the results of RT–RPA–CRISPR/Cas12a–LFD and quantitative polymerase chain reaction (PCR), while reducing the total detection time to 50 minutes. This approach eliminated the need for advanced instrumentation owing to its simplified operational design, enabling field-deployable rapid detection capabilities that establish essential technical infrastructure for initiating timely GETV containment measures. This approach has broad application potential in the fields of food safety, clinical diagnostics, and environmental science.

## INTRODUCTION

Getah virus (GETV), a mosquito-borne alphavirus belonging to the family *Togaviridae*, was first identified in *Culex gelidus* mosquitoes in Malaysia in 1955 (*Nemoto et al., 2015*). GETV primarily infects livestock, particularly pigs and horses, causing reproductive disorders in swine, high mortality in newborn piglets, and fever and hindlimb edema in horses (*Lu et al., 2019*; *Yang et al., 2018*). In recent years, GETV has re-emerged in multiple Asian countries, with frequent outbreaks reported in Chinese swine populations. The transmission of GETV is closely linked to mosquito activity, particularly during warm seasons when mosquito breeding and activity intensify, facilitating viral spread (*Sam et al., 2022*). Furthermore, GETV infections have been detected in cattle, blue foxes, red pandas, and other animals (*Shi et al., 2019*; *Liu et al., 2019*; *Zhao et al., 2022*), and GETV-specific antibodies have been identified in human sera, suggesting a potential public health threat (*Li, Wang & Liang, 2022*).

The GETV genome is a single-stranded positive-sense RNA of approximately 11.5 kb length. It encodes nine viral proteins, including four nonstructural proteins (nsP1–4) and five structural proteins (E1, E2, E3, 6K, and C). Among these, *Sun et al. (2022)* developed an enzyme-linked immunosorbent assay (ELISA) targeting the E2 protein of GETV, demonstrating its utility as a diagnostic antigen for the specific and sensitive detection of GETV antibodies in serological samples. Therefore, the E2 protein serves as a valuable diagnostic antigen for developing specific and sensitive detection assays.

Recombinase polymerase amplification (RPA), which does not require a complex thermal cycler but can be performed at a constant temperature using a simple water bath or heating block, has been shown to be a rapid, specific, sensitive and cost-effective technique for pathogen detection. The entire reaction can be completed in a short period of time (20 min) at a constant temperature (preferably 37 °C–42 °C), which makes RPA a promising field detection method (*Tan et al., 2022*). Also, RPA has been used to detect a wide range of pathogens such as bacteria, parasites and viruses (*Zhang et al., 2024*; *Bian et al., 2022*; *Mei et al., 2023*).

Clustered regularly interspaced short palindromic repeats (CRISPR) and CRISPR-associated (Cas) systems, which were first identified in the 1980s as a prokaryotic adaptive immune mechanism (*Cong et al., 2013*), have evolved into revolutionary tools for gene editing and molecular diagnostics (*Doudna & Charpentier, 2014*; *Samanta et al., 2022*). CRISPR-based immunity utilizes programmable RNA components cursor CRISPR RNA (crRNA) that spatially orient Cas nucleases for site-specific cleavage of exogenous nucleic acid targets, particularly those derived from viral pathogens or bacteriophage infections. These systems can be classified into two categories: Class 1 systems require multisubunit

Cas complexes, whereas Class 2 systems (*e.g.*, Cas12, Cas13, and Cas14) utilize a single effector protein for target recognition and cleavage (*Jinek et al., 2012*; *Makarova et al., 2015*). Notably, Cas12a (Cpf1), upon crRNA-guided recognition of a protospacer adjacent motif in double-stranded DNA (dsDNA), exhibits both cis-cleavage of the target dsDNA and trans-cleavage activity to nonspecifically degrade single-stranded DNA (ssDNA) (*Collias & Beisel, 2021*; *Chen et al., 2018*). Similarly, Cas13a—an RNA-targeting RNase— triggers the trans-cleavage of ssRNA upon binding to its target RNA (*Gootenberg et al., 2017*). Leveraging these properties, the CRISPR/Cas systems have been widely applied for pathogen detection. For instance, the Specific High-sensitivity Enzymatic Reporter unLOCKing (SHERLOCK) platform employs Cas13a to detect HBV, ZIKV, and SARS-CoV-2 (*Myhrvold et al., 2018*; *Joung et al., 2020*), whereas the Cas12a-based systems are preferred for DNA virus detection owing to their direct targeting of DNA without requiring any transcription. A recent study integrated Cas13a with lateral flow dipsticks (LFDs) to establish rapid, highly sensitive, and user-friendly diagnostic methods for the H5-subtype highly pathogenic avian influenza virus (HPAIV) (*Li et al., 2023*). This approach combines CRISPR/Cas13a-mediated RNA recognition and signal amplification with LFD-based visual readouts.

In this study, we developed a rapid, ultrasensitive GETV detection assay by integrating reverse transcription-recombinase polymerase amplification (RT-RPA), CRISPR/Cas12a, and LFD, targeting the conserved E2 gene. The employed method demonstrated high specificity and sensitivity, which was validated using simulated clinical samples. By eliminating the need for complex instrumentation, this approach significantly reduces costs while enhancing detection efficiency through RT-RPA-CRISPR synergy. The developed methodology establishes a rapid-response diagnostic framework for GETV surveillance, thereby delivering operational capabilities essential for coordinated veterinary–epidemiological containment strategies during emerging outbreaks.

## MATERIALS & METHODS

### Viruses and clinical samples

Getah virus (GETV), Japanese encephalitis virus (JEV), pseudorabies virus (PRV), porcine circovirus (PCV), and classical swine fever virus (CSFV) sera were provided by the Military Veterinary Research Institute, Academy of Military Medical Sciences (Changchun, China). For clinical validation, 30 porcine serum samples were collected from a swine farm affiliated with the Jinzhou Medical University in Liaoning Province (Experimental Animal Ethics Committee of Jinzhou Medical University Approval no. 2024176-5.

### Design of primers, crRNAs, and ssDNA reporters

The E2 protein gene sequences of GETV (NCBI Gene IDs: EU015063, EF631998, EU015061, EU015062, KY434327, KY450683, and MG869691) were aligned using MEGA 6.06, selecting the conserved region of GETV (EU015063) E2 protein from 179 bp–479 bp as the target gene to design two crRNAs, each paired with two primers. Two types of

**Table 1** The sequences of primers, probes and crRNA.

| Name | Sequence (5′–3′) |
|---|---|
| GETV-PCR-F | GAACACACGAACACAACAAAATCAGGTACAT |
| GETV-PCR-R | GTAGTCAGCTGGTACGTTGTGCATGGCACTT |
| crRNA A | UAAUUUCUACUAAGUGUAGAUUACUGCACUUUGCAGGCCUGG |
| crRNA B | UAAUUUCUACUAAGUGUAGAUUCUGCCUACUGGUGCCGGUGC |
| GETV-RPA-F1 | TTCGATACCGAAGTGGGGCCTGACGGTGAA |
| GETV-RPA-F2 | CATGGCACTTCGATACCGAAGTGGGGCCTG |
| GETV-RPA-R A1 | AACTAAAGGTCCAGTTCCAAGATGCAGAAT |
| GETV-RPA-R A2 | GGGGACGAACTAAAGGTCCAGTTCCAAGAT |
| GETV-RPA-R B1 | CACACCCAGGCCTGCAAAGTGCAGTACAAA |
| GETV-RPA-R B2 | ATGCAGAATCGCACACCCAGGCCTGCAAAG |
| DWV-FBD-ssDNA reporter | FAM-TTATT- BHQ |
| DWV-LFD-ssDNA reporter | FAM-TTTTTTATTTTTT-Biotin |
| GETV-RT-qPCR-F | AGCATTTTCGCATCTGGCTAC |
| GETV-RT-qPCR-R | TCTGGGTCTTCCGCACTTTT |

ssDNA reporters were synthesized: a fluorescence reporter (5′-FAM-TTATT-BHQ1-3′) for CRISPR-Cas12a-based fluorescence detection and an LFD reporter (5′-FAM-TTTTTTATTTTTT-Biotin-3′) for LFD-based visualization. The reverse transcription qualitative polymerase chain reaction (RT-qPCR) primers used to simulate the clinical assay were based on the GETV RT-qPCR assay established by *Cao et al. (2022)*. All primers, crRNAs, and ssDNA reporters (Table 1) were synthesized by Sangon Biotech (Shanghai, China).

## Construction of standard plasmid

First, extract RNA from the serum containing GETV using the TIANamp Virus DNA/RNA Kit (Tiangen, Beijing, China), and reverse transcribe the RNA into cDNA to use the cDNA of GETV as a template for PCR amplification. The polymerase chain reaction (PCR) system was 50 µL (50 µL reaction mixture: 25 µL 2× Taq Mix (Seven Biotech), two µL GETV-PCR upstream and downstream primers (10 µM) (Table 1), two µL template, 21 µL ddH2O) under the following conditions: 94 °C for 2 min; 35 cycles of 94 °C (45 s), 56 °C (45 s), 72 °C (45 s) and final extension at 72 °C for 10 min. The PCR products were verified *via* 1.2% agarose gel electrophoresis, purified using the SanPrep DNA Gel Extraction Kit (Sangon Biotech), and cloned into the pEASY-T1 vector (TransGen Biotech, Beijing, China). Recombinant plasmids were transformed into DH5α-competent cells, single white colonies were screened after overnight incubation using solid medium containing kanamycin sulphate, and the white colonies were selected and plasmids were extracted using the Plasmid Extraction Kit (Tiangen Biochemical Technology Co., Ltd.). The extracted standard plasmids were sequenced and verified by Sangon Biotech. The plasmid concentration was measured by a micro UV-Vis spectrophotometer (Thermo

Fisher Scientific, Waltham, MA, USA), and the copy number per unit volume of plasmid was calculated by formula and stored at −20 °C:

Plasmid copy number (copies/μL)=[plasmid concentation (g/μL)×10-9×6.02×1023]/{[Vector length (bp) + fragment length (bp)]×660/gmol}.

## Establishment of RT-RPA assay

RT-RPA reactions were performed using the commercially available RT-RPA Nucleic Acid Amplification Kit (Gendx Biotech, Suzhou, China) according to the manufacturer's specifications. In brief, 50 μL of reaction mixture contained 20 μL of rehydration buffer, 2.5 μL of each primer (10 μM), three μL of template, 20 μL of ddH2O, and two μL of the activator(magnesium acetate). A 48 μl premix of the reactants except activator was first prepared, and then the premix was transferred to a tube containing lyophilised enzyme RT-basic amplification reagent. After shaking and mixing, two μl of activator was added to the cap of the reaction tube, and the activator was briefly centrifuged into the premix. Finally, the reaction tube was incubated in a thermostat at 42 °C for 20 min. The amplified product (20 μL) was mixed with 40 μL of phenol, centrifuged, and screened for the best RT-RPA primers by 1.2% agarose gel electrophoresis.

## Fluorescence-based RT-RPA-CRISPR/Cas12a detection (FBDA)

The CRISPR-Cas12a fluorescence assay (total 20 μL) included one μL of crRNA (10 μM), two μL of 10× HOLMES buffer (TOLOBIO, Shanghai, China), 0.5 μL of LbCas12a nuclease (10 μM; TOLOBIO, Shanghai, China), one μL of ssDNA reporter (10 μM), 13 μL of nuclease-free water, and 2.5 μL of RPA product. The reaction mixture was incubated with QuantStudio1 (ABI, Natick, MA, United States) at 37.5 °C for 20 min, and the fluorescence (FAM) signal was detected every 20 s. The reaction tubes can be placed under an LED UV transmission unit (Tanon, Shanghai, China) to observe the fluorescence intensity with the naked eye after the test. Screening crRNA by FBDA

## RT-RPA-CRISPR/Cas12a-LFD assay

For lateral flow detection, the CRISPR-Cas12a reaction mixture was diluted to 50-μL volume with nuclease-free water and applied to an LFD strip (Gendx Biotech, Suzhou, China) (results appear in 3 min and are valid within 10 min). The results were interpreted as follows: dual red bands (control line (C) and test line (T)) indicated positivity; a single C band indicated negativity. Strips lacking a C band were deemed invalid and retested.

## Sensitivity and specificity testing

The constructed plasmids were diluted to $10^6$–$10^0$copies/μL and then tested for sensitivity using FBDA and RT-RPA-CRISPR/Cas12a-LFD assays, respectively, and the fluorescence intensity could be observed visually by placing the reaction tubes under the LED UV projection after the FBDA assay. The specific detection method was consistent with the sensitivity detection method, but the samples selected for the specific detection were nucleic acids extracted from viral serum (GETV, JEV, PRV, PCV and CSFV), and the DNA of PRV and PCV, as well as the RNA of GETV, JEV and CSFV, were extracted using the TIANamp Viral DNA/RNA Kit (Tiangen, Beijing, China), and the RNA was reverse transcribed into

cDNA. Nucleic acid concentrations were measured by a micro-UV-Vis spectrophotometer (Thermo Fisher Scientific) and the five nucleic acids were diluted to the same concentration of 50 ng/μL. All of the above assays were repeated three times to ensure the feasibility and reproducibility of the experiment.

### Validation with simulated clinical samples

No cases of infection by GETV have been reported in Liaoning province, China. Due to the scarcity of true positive samples, this study used a simulated positive sample strategy to comparatively analyse the accuracy between RT-RPA-CRISPR/Cas12a-LFD and reverse transcription qualitative polymerase chain reaction (RT-qPCR). Six tubes of serum containing GETV were randomly and blindly mixed into 30 clinical samples to form a sample containing 36 cases. All samples were completed by independent operators with unknown grouping information. All samples were subjected to RT-qPCR according to the method of *Cao et al. (2022)* for comparison with the results of RT-RPA-CRISPR/Cas12a-LFD assay.

## RESULTS

### Successful construction of an RT-RPA-CRISPR/Cas12a system for GETV detection (Fig. 1)

The CRISPR-Cas12a detection architecture integrates recombinant LbCas12a nuclease, CRISPR array-optimized crRNA guides, fluorogenic ssDNA reporters, and RPA-amplified dsDNA targets. Leveraging the programmability of CRISPR/Cas12a, crRNA was designed to base-pair with the conserved E2 gene of GETV. The integration of RT-RPA with CRISPR/Cas12a enhanced the detection sensitivity. In this system, preincubated Cas12a–crRNA complexes specifically recognize and cleave target dsDNA amplified *via* RT-RPA, activating the trans-cleavage activity of Cas12a to indiscriminately degrade ssDNA probes. Fluorescently quenched ssDNA probes served as reporters, and target-induced cleavage restored the fluorescence signals for quantitative detection. Additionally, an LFD assay was developed using FAM- and biotin-modified single-stranded DNA (ssDNA) reporters. In the negative samples, anti-FAM antibodies immobilized on the control (C) line captured uncleaved FAM-biotin reporters bound to gold nanoparticle-conjugated streptavidin. For the positive samples, the cleavage of reporters reduced the C-line intensity while enabling the accumulation of gold–streptavidin complexes at the test (T) line. The entire LFD readout was completed within 10 min, allowing for visual interpretation in field applications.

### Screening of the RT-RPA primers and crRNA

Optimal RT-RPA primers and crRNA were selected through agarose gel electrophoresis and RT-RPA–CRISPR/Cas12a fluorescence assays. According to the brightness of the target band and the amplification efficiency of the fluorescence amplification curve in the agarose gel electrophoresis image, the brightest band of interest in the agarose gel electrophoresis image (Fig. 2A) was selected as the best RT-RPA primer (GETV-RPA-F1 and GETV-RPA-R A1), and the group with the highest amplification efficiency of the
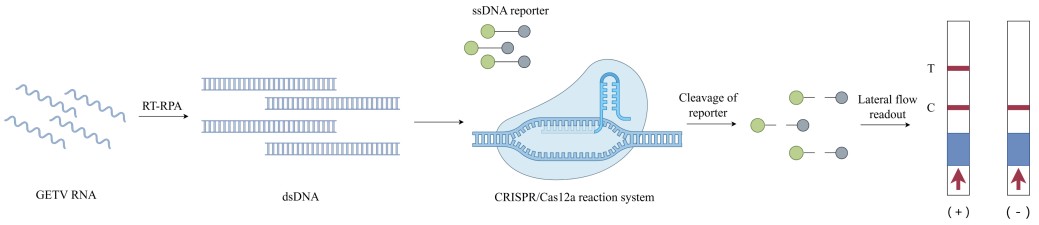

**Figure 1** Schematic diagram of the RT-RPA-CRISPR/Cas12a-LFD workflow for GETV detection.
Image credit: Figdraw.

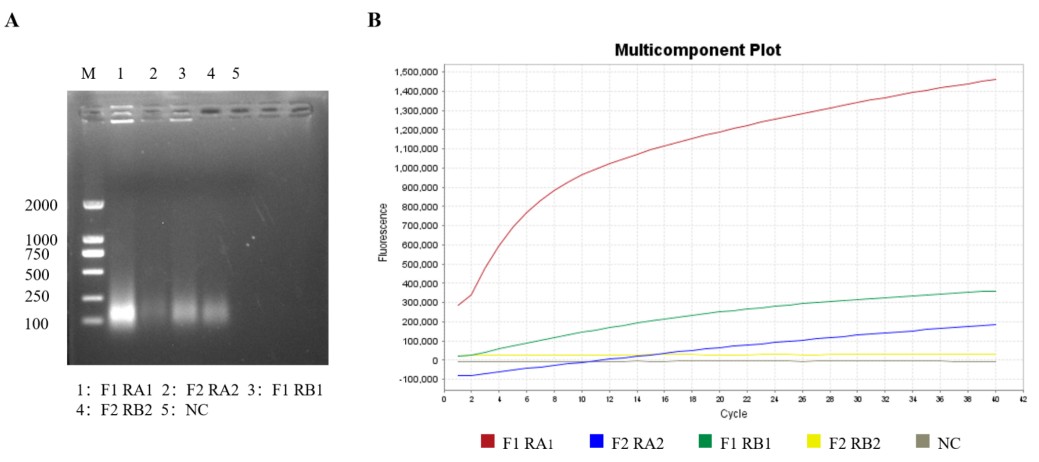

**Figure 2** Screening of primers and crRNA for the RT-RPA-CRISPR/Cas12a-LFD system. (A) Screening of optimal primers using RT-RPA. (B) Screening of optimal crRNAs based on RT-RPA-CRISPR/Cas12a real-time fluorescence.

fluorescence amplification curve (Fig. 2B) was selected as the best crRNA (crRNA A), that is, crRNA A was selected and the first pair of primers (GETV-RPA-F1 and GETV-RPA-R A1).

## Sensitivity of the RT-RPA-CRISPR/Cas12a system

The sensitivity of the RT-RPA-CRISPR/Cas12a system was evaluated using serially diluted GETV standard plasmids ($10^6$–$10^0$ copies/$\mu$L) and nuclease-free water as a negative control. Following RPA amplification, CRISPR/Cas12a-mediated fluorescence detection achieved a limit of detection of $10^1$ copies/$\mu$L across triplicate reactions (Fig. 3A). Parallel testing with the LFD confirmed visible T-line signals at $10^1$ copies/$\mu$L (Fig. 3B). Moreover, the detection sensitivity was repeated three times by LFD method, which proved the reproducibility of the experiment (Fig. 1S), validating the system's high sensitivity.

## Specificity of the RT-RPA-CRISPR/Cas12a system

The specificity of the assay was validated against four common swine pathogens: JEV, PRV, PCV, and CSFV. Both the fluorescence-based and LFD assays showed no cross-reactivity (Figs. 4A and 4B) (Fig. 2S), confirming the method's high specificity for GETV.

Peer J

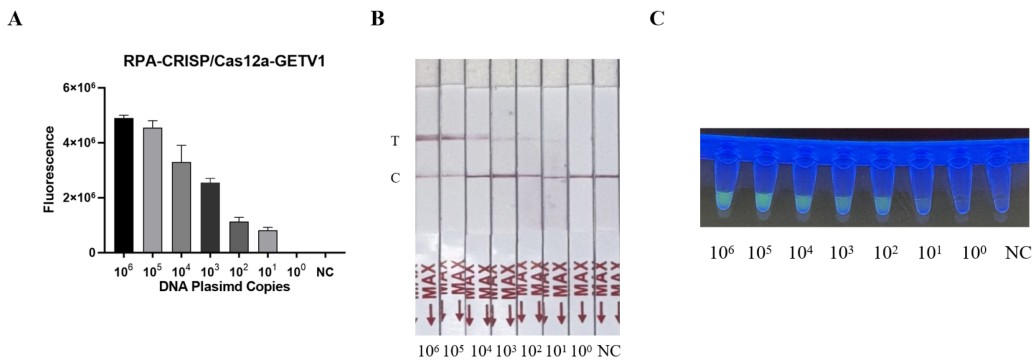

**Figure 3** **Sensitivity assessment of the RT-RPA-CRISPR/Cas12a system coupled with lateral flow detection.** (A) Histogram of GETV detection limits by RT-RPA-CRISPR/Cas12a-FBDA. (B) Analytical sensitivity of RT-RPA-CRISPR/Cas12a-LFD in GETV detection. (C) Detection using a UV transilluminator allowed examination with the naked eye.

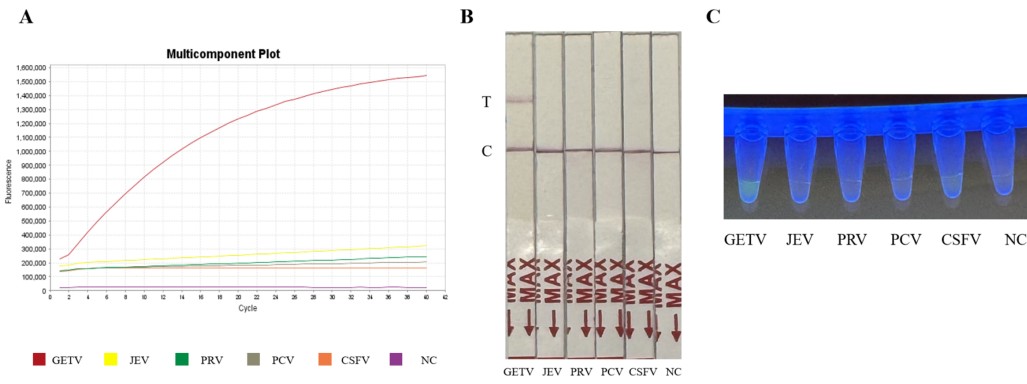

**Figure 4** **Specificity assessment of the RT-RPA-CRISPR/Cas12a assay.** (A) Specificity of the GETV RT-RPA-CRISPR/Cas12a-FBDA assay. (B) Specificity of the GETV RT-RPA-CRISPR/Cas12a-LFD assay. No cross-reactivity with JEV, PRV, PCV or CSFV was observed. (C) Observation of fluorescence intensity using ultraviolet transmission illuminator test.

## Detection of simulated clinical samples

To assess clinical utility, 36 simulated samples were analyzed using the RT-RPA-CRISPR/Cas12a-LFD assay and compared with RT-qPCR. Both methods identified six positive samples with 100% concordance (Fig. 5, Table 2). The RT-RPA-CRISPR/Cas12a-LFD assay completed detection within 50 min.

## DISCUSSION

As a mosquito-borne alphavirus, GETV is increasingly escalating economic losses in the livestock industry across the Asia-Pacific region by infecting pigs and horses and posing potential zoonotic risks to humans (*Yang et al., 2018*; *Rattanatumhi et al., 2022*). To address the limitations of the current detection methods, such as insufficient sensitivity, equipment dependency, and field applicability, we successfully established a rapid GETV detection
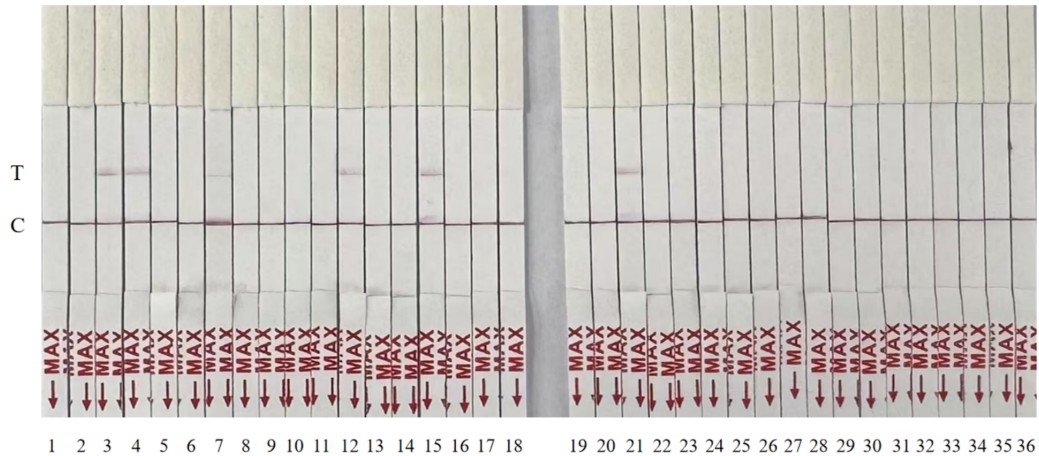

**Figure 5** Randomized testing of 36 simulated clinical samples with the RT-RPA-CRISPR/Cas12a-LFD system.

platform (RT-RPA–CRISPR/Cas12a–LFD) by integrating RT-RPA with CRISPR/Cas12a. This platform achieved a detection limit of $10^1$ copies/µL and completed the entire process within 50 min, thus providing critical technical support for early outbreak containment.

The enhanced performance of this platform stems from the following three synergistic innovations: (1) the crRNA (*i.e.,* crRNA A) targeting the conserved E2 gene of GETV demonstrated exceptional specificity by effectively distinguishing GETV from cocirculating pathogens, such as JEV and PRV (*Zhang et al., 2022*); (2) The optimized RT-RPA system enables nucleic acid amplification in less than 20 min at 37−42 °C using a simple heating device; (3) the trans-cleavage activity of Cas12a, coupled with a dual-labeled (FAM/biotin) ssDNA reporter system (*Rananaware et al., 2023*), allowed visual interpretation *via* LFDs within 3 min. Clinical validation revealed 100% concordance with the RT-qPCR results.

Currently there are many methods regarding the detection of GETV such as RT-qPCR (*Cao et al., 2022*), RT-RAA (*Nie et al., 2021*), RT-PCR and ELISA (*You et al., 2024*). PT-qPCR, RT-PCR and RT-RAA require expensive instrumentation and these methods have poor field applicability compared to existing GETV detection techniques, and for ELISA assays, the sensitivity is only $10^3$ copies/µL, which is insufficient to cover the need for detection in the early stages of viraemia (24–48 h post-infection) (*Shi et al., 2022*). The amount of GETV in the serum of infected animals is approximately $7.73 \times 10^5$ copies/µL (*Cao et al., 2022*), and the RT-RPA-CRISPR/Cas12a-LFD constructed here is sufficient to detect the Geta virus. In summary, the use of a combination of RT-RPA and LFD in our established CRISPR-Cas12a fluorescence detection system enables rapid visual detection of GETV. The method is simple to perform, does not require expensive equipment, provides more intuitive reaction results, has strong specificity and high sensitivity, and can be performed at a constant temperature or even at body temperature using a simple water bath or heating block. The method provides early warning of GETV transmission. The method can also be used for the detection of other pathogens. The method has promising applications in food safety, clinical diagnostics and environmental science.

**Table 2   Detection of GETV in simulated clinical samples using the RPA-CRISPR-Cas12aLFD assay and comparison with RT-qPCR performance.**

| | Assay results | |
|---|---|---|
| Sample ID | RT–qPCR | RPA–CRISPR–Cas12a–LFD |
| 1 | − | − |
| 2 | − | − |
| 3 | 18.432 | + |
| 4 | 21.678 | + |
| 5 | − | − |
| 6 | − | − |
| 7 | 29.366 | + |
| 8 | − | − |
| 9 | − | − |
| 10 | − | − |
| 11 | − | − |
| 12 | 23.214 | + |
| 13 | − | − |
| 14 | − | − |
| 15 | 18.446 | + |
| 16 | − | − |
| 17 | − | − |
| 18 | − | − |
| 19 | − | − |
| 20 | − | − |
| 21 | 18.742 | + |
| 22 | − | − |
| 23 | − | − |
| 24 | − | − |
| 25 | − | − |
| 26 | − | − |
| 27 | − | − |
| 28 | − | − |
| 29 | − | − |
| 30 | − | − |
| 31 | − | − |
| 32 | − | − |
| 33 | − | − |
| 34 | − | − |
| 35 | − | − |
| 36 | − | − |

Notes.
Ct,  mean threshold cycle value of positive samples;  +,  positive detection;  −,  negative detection.

Although the method shows promising applications, clinical sample pretreatment still relies on column nucleic acid extraction, and extraction-free direct amplification technology can be developed in the future to further simplify the process.

## CONCLUSIONS

In conclusion, the integration of RT-RPA, CRISPR/Cas12a, and LFD into our system enables rapid, special equipment-free, and visually interpretable GETV detection. With the advantages of high specificity, sensitivity, and compatibility with simple heating devices (*e.g.*, water baths and heating blocks), this method can act as an early warning tool for GETV transmission and hold broad potential for adaptation to other pathogens in fields such as food safety, clinical diagnostics, and environmental monitoring.

### Funding

This work was supported by the National Key R&D Program of China (2023YFD1800404), the Science and Technology Program of Liaoning Province (2024-MS-238), the Project of the Department of Education of Liaoning Province (LJ212410160093), and the Scientific Research Fund of the First Affiliated Hospital of Jinzhou Medical University (Grant KYTD-2022004). The funders had no role in study design, data collection and analysis, decision to publish, or preparation of the manuscript.

### Grant Disclosures

The following grant information was disclosed by the authors:
The National Key R&D Program of China: 2023YFD1800404.
The Science and Technology Program of Liaoning Province: 2024-MS-238.
The Project of the Department of Education of Liaoning Province: LJ212410160093.
The Scientific Research Fund of the First Affiliated Hospital of Jinzhou Medical University: KYTD-2022004.

### Competing Interests

Xiuwei Shu is employed by Liaoning Yikang Biological Co., Ltd.

### Author Contributions

- Boyang Xia performed the experiments, analyzed the data, prepared figures and/or tables, and approved the final draft.
- Ziyan Wang performed the experiments, prepared figures and/or tables, and approved the final draft.
- Tiantian Fei analyzed the data, prepared figures and/or tables, and approved the final draft.
- Yueyu Ma analyzed the data, prepared figures and/or tables, and approved the final draft.
- Yaxi Guo analyzed the data, prepared figures and/or tables, and approved the final draft.

- Dongliang Fei conceived and designed the experiments, authored or reviewed drafts of the article, and approved the final draft.
- Xiuwei Shu conceived and designed the experiments, authored or reviewed drafts of the article, and approved the final draft.
- Gang Zhao conceived and designed the experiments, authored or reviewed drafts of the article, and approved the final draft.
- Mingxiao Ma conceived and designed the experiments, authored or reviewed drafts of the article, and approved the final draft.
- Hongxia Yuan conceived and designed the experiments, authored or reviewed drafts of the article, and approved the final draft.

### Animal Ethics

The following information was supplied relating to ethical approvals (*i.e.*, approving body and any reference numbers):

All animal experiments described in this study were conducted in strict accordance with the guidelines and regulations for the care and use of laboratory animals. The experimental protocols were reviewed and approved by the Experimental Animal Ethics Committee of Jinzhou Medical University (Approval No. 2024176-5).

### Data Availability

Data is available at NCBI: EU015063, EF631998, EU015061, EU015062, KY434327, KY450683, and MG869691.

Raw data are available as Supplemental Files.

### Supplemental Information

Supplemental information for this article can be found online at http://dx.doi.org/10.7717/peerj.20119#supplemental-information.

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
