# Peer review of "Development and application of a CRISPR/Cas12a-based reverse transcription–recombinase polymerase amplification assay with lateral flow dipstick and fluorescence detection for Getah virus"

_PeerJ, doi:10.7717/peerj.20119_

## Round 0.1 · original submission · Major Revisions

· Academic Editor

Major Revisions

While the study addresses an important topic and shows promise, the reviewers and I have identified several critical issues that must be addressed. These include the lack of a clearly stated objective in the introduction, major language and structural issues throughout the manuscript, and insufficient methodological detail that currently limits reproducibility. Figures are of inadequate quality and require significant improvement, with several being unclear or missing proper labels and references. Additionally, several claims, particularly around assay sensitivity, are not fully supported by the data, and further clarification is needed regarding spiked sample preparation and LFD assay replication. Regarding the analytical sensitivity testing using LFD, if the results presented were based on a single replicate, we would expect you to repeat the LFD testing (as done for fluorescence detection) to confirm reproducibility, especially at lower concentrations. The current data, particularly the faint bands, require further validation to support reliable detection claims. Depending on your response and additional data provided, the extent of revision required may remain major. Ethical approval for the use of animal samples should also be clearly stated. Please respond thoroughly to each reviewer comment, revise the manuscript accordingly, and highlight all changes.

I look forward to receiving your revised submission

Reviewer 1 ·

Basic reporting

This manuscript presents a study utilizing a CRISPR/Cas12a-based reverse transcription-recombinase polymerase amplification (RT-RPA) assay for the diagnosis of Getah virus, incorporating both lateral flow dipstick and fluorescence detection methods. Overall, the scientific rigor and presentation of the manuscript require substantial improvement. A comprehensive revision and reorganization of all sections is necessary to elevate the academic quality. Furthermore, the figures currently fall below acceptable standards for scientific publication and will need significant enhancement to effectively convey the study's findings.

Experimental design

Introduction:
The introduction omits a crucial element: a clear statement of the study objectives. Consequently, the scientific gap that the manuscript aims to fill remains unclear. The introduction requires significant revision to improve its clarity, focus, and effectiveness.
1- Line 56: Please, put references 4 in brackets.
2- Line 63: Please, remove a ( i.e.) and clarify the exact number of structural proteins in the virus, rather than using an approximation..
3- Lines 63-68: the section " Among these, the E2 protein is a key glycoprotein ….." it is appears to be extraneous to the primary objective of the introduction. Its removal is recommended to enhance the conciseness and focus of the introductory material.
4- Table 1, please revise the legend for Table 1. Specifically, the term "probes" should be replaced with "reporters".
5- Line 122: Authors should include the names of the primers used in the PCR.
6- Line 135: Please clarify what the activator is and put the concentration as µl.
7- Line 149, please, remove a statement (as mentioned above).

Validity of the findings

Results

The results section exhibits significant weaknesses in its current form. There are grammatical and spelling errors prevalent throughout. Some findings lack scientific rigor and appear to be overstated. The presentation of molecular examination results is ambiguous and requires further clarification. The virus isolation section is deemed irrelevant and doesn't support the results of the manuscript, and should be removed.
1- Line 162: section 1. Schematic of the RT-RPA.CRISPR/Cas12a System for GETV Detection, is requires a brief rephrasing.
2- How authors optimize the quantity of reporters in samples and reduce the false positives.
3- Line 179: section 2. Screening of the RPA Primers and crRNA. The authors should delete this section because it is unclear, and the results of fluorescence amplification curves are wrong.
4- Figure 2A and B are unclear and don't support the result, particularly fluorescence amplification curves; please, delete them.
5- Figure 3C is not reported in the manuscript text.

Discussion
The discussion section should be revised to ensure it is professionally written and directly relevant to the study's findings after revision. They put the limitations of the research and discussed them transparently.

Conclusion
A concise and impactful conclusion should be formulated. This conclusion should accurately summarize the key results extracted from the manuscript and their significance.

Additional comments

The figures currently fall below acceptable standards for scientific publication and will need significant enhancement to effectively convey the study's findings.

Reviewer 2 ·

Basic reporting

Please find some general comments about the document overall and the introduction and figures/tables in particular below.

Thank you for the short introduction to the Getah Virus and the CRISPR-Cas system. While some of the additional information provided is interesting (e.g., about Cas13a), it is not directly relevant in the context of the study. I recommend focusing on the introduction of the actual Cas (Cas12a) used in this study rather than interspersing it with Cas13a references. Also, as this study focuses on an isothermal amplification method, the addition of information about the use of isothermal amplification in the detection of GETV would make the introduction more relevant in the context of the study. For instance, Cas12a has been used in other studies in combination with RPA (see Low SJ, O’Neill MT, Kerry WJ et al, 2023; Rapid detection of monkeypox virus using a CRISPR-Cas12a mediated assay: a laboratory validation and evaluation study; Lancet Microbe as an example). Also, isothermal amplification has previously been used for Getah virus detection (e.g., Nie M, Deng H, Zhou Y et al. 2021, Development of a reverse transcription recombinase-aided amplification assay for detection of Getah virus, Scientific Reports [using RAA in this study] and Liu H, Li L-X, Bu Y-P et al 2019, Rapid visual detection of Getah virus using a loop-mediated isothermal amplification method, Vector-Borne and Zoonotic Diseases, 19(10)).

Please ensure high-quality figures are provided. Currently, Figure 1 is quite blurry and difficult to read.

While it is great to see lateral flow strip results, recommend indicating control and test lines for all RPA figures, particularly Figure 5 to increase clarity for readers not as familiar with lateral flow strips (or familiar with a different type of lateral flow strip that shows the control line on the top and the test line on the bottom).

Please address the following referencing issues:
- Recommend rewording and reviewing the references used for the CRISPR-Cas system explanation in the introduction, particularly Lines 76-78. The associated references only mention class 2 (types II and V) or three types of CRISPR/Cas systems (rather than the two mentioned by the authors). The following two example references are discussing Class 1 and Class 2 CRISPR-Cas systems, with Class 1 (types I, III, and IV) possessing multisubunit crRNA-effector complexes and Class 2 (types II, V, and VI) a single-subunit one. References:
o Chaudhary E, Chaudhary A, Sharma S, Tiwari V & Garg M (2024). Different Classes of CRISPR-Cas Systems, Chapter in Gene Editing in Plants, pp 73-94.
o Makarova KS, Wolf YI, Alkhnbashi OS et al. (2015). An updated evolutionary classification of CRISPR-Cas systems. Nature Reviews Microbiology 13, 722-736.
- Line 56 – reference 4 -> [4]
- Some of the references in the reference list contain “Published…” which does not appear to be a requirement/optional addition based on the author guidelines, and also makes the reference list inconsistent in its presentation.

Please review the document and carry out editing (grammar, formatting, abbreviations, and so on). A few examples (not comprehensive) are given below:
- appropriate species formatting – e.g., Line 50 Togaviridae, Line 51 Culex gelidus, which should be italicized.
- spacing and punctuation (e.g., Line 116, 117, Figure 2 legend)
- superscript – e.g., Line 129 (106-100 copies/µL)
- grammar/spelling, e.g., Line 67 “herefore” after the full stop is presumably “Therefore” (capitalized) and Figure 3A x-axis label.
- Please write out abbreviations – e.g., Line 74, and abbreviations in the figures and tables, e.g., NC, T, C, DWV
- Some of the wording is awkward and not as clear as it could be, e.g., lines 116-117. Stating that a previously established GETV RT-qPCR assay by Cao was used might clarify what the authors have done in this study.
- Sentence fragment – e.g., Lines 134-135
- Unusual wording in the context of the study, e.g., “synergy” (Line 97), “architecture” (Line 164)
- Please review your use of qPCR. qPCR and RT-qPCR are not the same, and it is important to use the correct terminology throughout your manuscript text.

Line 8 and line 18 – please indicate what * stands for (assuming this highlights the corresponding authors?).

Experimental design

As it currently stands, the methods section is missing some information that would allow replication of the study. Please revise the methods section to ensure sufficient detail is provided for each block of work you have carried out. Some examples are provided below:

- Considering serum samples were collected from pigs, I would expect to see some kind of ethics statement/statement about which committee approved animal use in this study. I could not find a statement in this regard in the manuscript document or the supplementary data files. Please disregard if this has been provided in a section reviewers have no access to – if it has not been provided anywhere else, please ensure this is included in your manuscript.
- Lines 120 - 121 – could the authors please add which kit was used for the extraction and cDNA synthesis
- Line 123 – could the authors please add the name of the mastermix used and the manufacturer
- Line 122 – Are the GETV-specific primers listed in the primer Table? If they are not, could the authors please add them?
- Line 128 – The authors mention “screening”. Please indicate what the screening method was.
- Lines 129-130 – Please include how the copy number was determined. Also, it is unusual to prepare a full serial dilution (particularly low dilutions) and then store it instead of using it straight away. This can be problematic, particularly for low copy number dilutions, especially if multiple freeze-thaws are carried out on the samples. Generally, small aliquots of high copy number dilutions are frozen to limit freeze-thawing, and full serial dilutions are carried out directly prior to use. Could the authors please provide reasoning for their method?
- Lines 119-130 – Could the authors please provide information on what the plasmids were used for? Generally, the methods section includes an “analytical sensitivity [and specificity]” section, which is missing here. Assuming the plasmids are used for analytical sensitivity testing, could this please be included in the methods section? Based on the raw data and figure(s), the plasmid seems to have been tested in triplicate with fluorescence (3A) and once each for 3B and 3C? Please clearly state in your methods what has been done, including how it was measured (e.g., how the fluorescence values in the raw data table were determined. There is also figure 4 that implies specificity testing was carried out with the other viruses? Recommend adding an “analytical sensitivity and specificity” section to your methods and clearly outlining there what has been done (including concentrations of non-GETV virus samples used for the analytical specificity testing).
- Lines 134-138 – Could the authors please revise their protocol? This is how it currently reads for a reader of the manuscript: a 50 µL reaction mixture was prepared (including 2 µL activator), then 48 µL of it was added to the RPA pellet, then another 2 µL of the activator was added to the rehydrated pellet. Based on other RPA reaction protocols, magnesium (the activator) is generally only added once, right at the end, to start the reaction, and it is unusual not to use the full master mix for pellet rehydration.
- Lines 140-147 – Could you please clarify if incubation and monitoring happened in the QuantStudio for 20 min? Was the HOLMES buffer also from TOLOBIO? Could you please include manufacturer details for the LED transmission fluorometer? Also, in the Figure 3C legend, it states a UV transilluminator (manufacturer?) was used to examine the result with the naked eye – this deviates from what is written in the methods. Could the authors please revise the methods section accordingly?
- Line 150: Please add information about the lateral flow strips used.
- Lines 156-157: Could the authors please explain how 36 simulated samples were prepared from spiking serum into 30 clinical sera? Also, could the authors please elaborate on how much of the serum (was this the GETV positive serum listed in section 1 of the method?) was added to how much of the clinical sample? When the authors say “using a blinded, randomized protocol,” does that mean different concentrations of the GETV-positive serum were spiked into the clinical sera, or that some were spiked, some were not? Could the authors please elaborate?
- Line 159 – What do the authors mean by “following the manufacturer’s protocols”? A RT-qPCR assay by Cao was mentioned in section 2 of the Methods; however, it does not seem to have been used so far. Was that assay used in section 7 of the Method instead of a commercial assay (as the statement implies)? If a different assay was used as a comparison/gold standard, could the authors please clarify what the assay by Cao (line 117) was used for?
- Generally, positive and negative (no-template controls or sample negatives) are included in testing. There is no mention of controls in the methods section. Based on the figures at the end of the manuscript, NC (assuming this stands for negative control) and positive controls are included for some of the testing, but not all? E.g., 4A has GETV and NC, but 4B and 4C only have GETV, and Figure 5 suggests no positive or negative controls were included in the testing.

Validity of the findings

In Lines 96-97 (“enhancing detection efficiency”) and line 168 (“significantly enhanced the detection sensitivity”), the authors are using quite strong language that is not supported by their data. The only comparison that was done was with? RT-qPCR? for the spiked clinical samples and based on the Table (Table 2), their platform did not perform any better (and certainly not “significantly” better) than the standard they are comparing to. The authors also have not carried out any testing with just RPA and compared it to RPA with CRISPR, so it is unclear what data/results these statements refer to.

Can the authors please review redundancy in the results section (e.g., Section 1)?

Can the authors please review their results section and ensure only results are presented? E.g.:
- Line 178 – authors state that the LFD readout was completed within 10 min, but in line 150, they state 3-5 min. If the results were not read right after incubation, but within a 10-minute window, this should be mentioned in the methods, not the results section.
- Lines 180-183 – how optimal primer combinations were chosen should be part of the methods, not results.

The authors have two sections (sections 2 and 3) with the same heading (“Screening of the RPA Primers and crRNA”). Could this either be rephrased or the two sections combined under the one subheading?

Line 187 – Please be aware that there are two types of sensitivity and specificity, analytical and clinical. Please make sure to clearly state that you had assessed analytical sensitivity with the plasmids and analytical specificity with the other viruses (Line 194).

Line 191 – Has the LFD analytical sensitivity testing been carried out more than once? Considering how faint the test bands are at 103-101 (particularly 101) in Figure 3B, it would be beneficial for the reader to see the results of replication, as really faint bands like these can drop off in replicate testing.

Lines 193-196 – as mentioned in the methods, the results presented here would be more meaningful if the concentrations of the four pathogens were known (please add this information to the methods section).

Lines 199 and 201 – could the authors please clarify if a qPCR or an RT-qPCR assay was used for this? Also, how long did the RT-qPCR (or qPCR as the authors used) take? Some PCR assays can be done in 1.5 hours with a lot fewer extra steps, so if the authors want to highlight time, that should be included – also, please be aware this is a results and not a discussion section, so this has to be appropriately phrased (also Line 202).

Lines 198-200 – While it is great to see that the newly developed test gives the same results as RT-qPCR, with the missing/unclear information in the methods section, it is a bit of an unusual result seeing that it currently reads like the GETV positive sample was spiked into all 30 samples (not sure about the extra 6) but was only detected in 6 out of 36 samples? In which case, both the RT-qPCR and the newly developed test are not particularly good in detecting the virus in actual samples unless it has a fairly high concentration (except for sample 7).

Some suggestions/concerns for the discussion section:
- There is very little discussion of the results in the context of the literature. For instance, what is the concentration of the virus generally found in real samples (important as the only samples tested are spiked samples, and currently there is no information as to the level at which they were spiked and whether that would be relevant to real samples or not)? This is not the first test that used, e.g., isothermal amplification methods for GETV testing. How does the test the authors have developed compare to those? Is serum the preferred way to test for this virus?
- Lines 207-208: Based on what the authors write in later sections of the discussion (line 225), there are more sensitive tests available.
- Lines 218-219: It is unclear why a single-tube reaction design is mentioned, as the authors have not used this design overall. Based on what was presented in the manuscript, the authors carried out a separate amplification of the target, then the tube was opened, 2.5 µL of the product pipetted out of the tube and into the CRISPR-Cas12 assay tube, followed by another incubation step.
- Line 225 – What is the analytical sensitivity for the qPCR (RT-qPCR) that is mentioned?
- Currently, spiked sample detection and the discussion appear to focus on LFD results, and it is unclear why the other two detection methods were included in this paper. Could the authors please provide some reasoning for the inclusion of the other detection methods?
 
Conclusion:
- Line 236: “equipment-free”. If this is something the authors want to highlight, this has to be discussed previously (at least the potential of it being equipment-free), as currently based on what is in the manuscript, several pieces of equipment are needed for the extraction and amplification steps (and only extraction has been mentioned in passing).

Additional comments

This manuscript describes the design and testing of a reverse transcription-recombinase polymerase amplification test for the detection of Getah virus in pigs that was coupled with CRISPR/Cas12a. Overall, the authors followed the standard sections outlined in the author guidelines and provided relevant raw data as per the journal’s data sharing policy. However, this manuscript could be written and presented more clearly. One example is the methods section, which is missing quite a bit of information, which makes some areas difficult to understand and assess for their relevance/importance.

---

## Round 0.2 · Minor Revisions

· Academic Editor

Minor Revisions

Thank you for submitting the revised version of your manuscript. I appreciate the considerable effort you have put into addressing the reviewers’ and editorial comments.

I have been unable to get the reviewers to re-review your revised manuscript, but I note several positive improvements in your revision.

The introduction now clearly states the study objective with improved focus and context. Methodological details have been clarified, including the listing of primers, kits, and manufacturers, and the addition of a section describing sensitivity and specificity testing. Ethical approval for the use of animal samples has been included. Figures have been improved, with clearer versions of Figures 2 and 5 and appropriate labels for control and test lines. Terminology and references have been corrected, and the use of overly strong claims has been moderated in the discussion.

Before I can proceed to acceptance, however, a few remaining issues must be addressed.

It remains unclear whether the LFD detection at low concentrations was performed in replicates. The figures still suggest single replicates with faint bands. Please clarify in the Methods section how many replicates were performed for each dilution in the LFD assay and ensure this is reflected in the Results. If only a single replicate was performed, additional replicates should be conducted to strengthen reproducibility.

Please also ensure that all statistical analyses meet expected standards: provide details on replicate numbers, consistency in reporting of n, exact p-values where applicable, and clarify whether any corrections for multiple testing were needed. At present, the statistical description is minimal.

In addition, please ensure that all figures are provided in high-quality, publication-ready format. Finally, although much improved, there are still instances where “qPCR” and “RT-qPCR” appear interchangeably. Please review the manuscript carefully and ensure consistent and correct terminology throughout.

Once these issues are addressed, the manuscript should be suitable for publication. I look forward to receiving your revised submission.

---

## Round 0.3 · accepted · Accept

· Academic Editor

Accept

I have carefully reviewed your revised manuscript along with your response letter. I am satisfied that you have addressed all reviewer and editorial comments. The revisions have improved the clarity and quality of the manuscript, and no further issues remain that require attention.

I am therefore pleased to inform you that your manuscript is accepted for publication in PeerJ.

Congratulations